# Identifying Chemical Differences in Cheddar Cheese Based on Maturity Level and Manufacturer Using Vibrational Spectroscopy and Chemometrics

**DOI:** 10.3390/molecules28248051

**Published:** 2023-12-12

**Authors:** Gerson R. Dewantier, Peter J. Torley, Ewan W. Blanch

**Affiliations:** 1Applied Chemistry and Environmental Science, School of Science, Royal Melbourne Institute of Technology University, Melbourne, VIC 3001, Australia; s3736548@student.rmit.edu.au; 2Biosciences and Food Technology, School of Science, Royal Melbourne Institute of Technology University, Bundoora, VIC 3083, Australia; peter.torley@rmit.edu.au

**Keywords:** cheese ripening, cheddar cheese, vibrational spectroscopy, chemometrics

## Abstract

Cheese is a nutritious dairy product and a valuable commodity. Internationally, cheddar cheese is produced and consumed in large quantities, and it is the main cheese variety that is exported from Australia. Despite its importance, the analytical methods to that are used to determine cheese quality rely on traditional approaches that require time, are invasive, and which involve potentially hazardous chemicals. In contrast, spectroscopic techniques can rapidly provide molecular information and are non-destructive, fast, and chemical-free methods. Combined with partner recognition methods (chemometrics), they can identify small changes in the composition or condition of cheeses. In this work, we combined FTIR and Raman spectroscopies with principal component analysis (PCA) to investigate the effects of aging in commercial cheddar cheeses. Changes in the amide I and II bands were the main spectral characteristics responsible for classifying commercial cheddar cheeses based on the ripening time and manufacturer using FTIR, and bands from lipids, including β’-polymorph of fat crystals, were more clearly determined through changes in the Raman spectra.

## 1. Introduction

Cheddar cheese is a well-known cheese variety, and it is widely produced and consumed around the world. It is also exported by various countries, including Australia and New Zealand, making it a significant commodity in international trade. The ingredients used in cheddar cheese include pasteurized cow’s milk, salt, lactic acid bacteria (LAB), and proteases. Traditionally, the protease is calf’s rennet; but, more recently, vegetable or bacterial chymosin have become common. Cheddar cheese is a rennet-coagulated cheese, that is salt-dried, hard (water content < 38%), and internal bacterially ripened [1]. Some sensorial attributes of cheddar cheese include salty, acid, bitter, and mouth coating, and the texture ranges from rubbery to crumbly depending on the maturation time [2,3]. Cheddar cheese is sold at different maturity levels from less than 2 months to more than 2 years—up to 10 years in rare instances. A characteristic step of its manufacture is the cheddaring process, which involves the stacking of blocks of curds on each other, causing deformation and fusion of the curd particles, leading to the characteristic final texture of cheddar cheese [3].

Despite its importance, the methods used to measure cheese parameters mostly rely on traditional techniques of analysis. For example, common methods used to assess the biochemical changes during cheese ripening include colorimetric or enzymatic chromatography and electrophoresis [4]. These methods are accurate but time-consuming; they require sample preparation, skilled operators, and may use hazardous chemicals. Food processors and regulatory agencies would prefer a technique that can be used to quickly screen complex food matrices like cheese. Vibrational spectroscopies, namely Fourier-transform infrared (FTIR) and Raman spectroscopies, are sensitive to large amounts of molecular information related to sample composition and have been successfully applied in many areas of food science, including characterizing alcoholic spirits [5], wine [6], beef [7,8], fish [9], and cheese [10,11].

When combined with chemometrics, vibrational spectroscopy becomes a powerful tool for quickly identifying various attributes in cheese, such as ripening [12], and for determining the levels of major components like fat, protein, and moisture [13]. Some minor components like free fatty acids (FFA) [14], pH [15], and free amino acids (FAA) [16] have also been reported. To date, few published studies have been conducted to identify the differences between commercial brands of cheese, and most of these have been concerned with demonstrating that the cheese comes from a particular Protected Designation of Origin (PDO) [17,18,19,20]. Lei et al., 2019, developed an approach of classifying commercial cheddar cheeses through combining near-infrared hyperspectral images and chemometrics, correctly classifying the cheeses with up to 86% accuracy [21].

To optimize the data analysis, researchers have applied different pre-processing methods, as well as testing different spectral ranges to enhance the differences between the samples in order to obtain the best accuracy. For example, Subramatrian et al. [12] used a range of only 900–1800 cm^−1^ in their FTIR studies on the ripeness of cheddar cheese. In the study of trans-fatty acids in dairy products, Zhao et al., 2015, tested nine different ranges of frequency of FTIR spectra, eight different ranges of frequency for NIR, and three different ranges of frequency for Raman spectra, to see which one would be the best fit for their prediction model. They found that the best-performing models were not constructed from the full range of the spectra [14]. Yaman et al. included the CH_2_/CH_3_ stretching mode bands from the FTIR spectra (3000–2800 cm^−1^) in their data analysis, but not the corresponding bands from their Raman spectra, to find which one worked best to verify the changes that occurred during the ripening of a brine-matured white Turkish cheese [22].

Our research aims to identify chemical changes in commercial cheddar cheese due to aging and due to manufacturer. Non-destructive techniques, FTIR and Raman, were applied, and the spectral variations were identified using principal component analysis (PCA). Ripening events, manufacturing conditions, and feedstock lead to different transformations in the cheeses. Note that, in this manuscript, we use the term ‘feedstock’ to refer to all the ingredients that play a role in cheddar maturation: milk, LAB, rennet, and salt.

## 2. Results and Discussion

### 2.1. Spectral Characterization of Commercial Cheddar Cheese

The infrared and Raman spectral bands were assigned with respect to published band allocations (Appendix A). Figure 1 shows an average of 840 spectra for both the FTIR and Raman spectra. An example of spectral variation has been included in Appendix A.

The main components in cheese are water, lipids, and proteins, and the specific bands for these compounds are readily observable in the FTIR spectra (Figure 1a) in the region between 400 and 4000 cm^−1^. A prominent band from water molecules is apparent between 3000 and 3600 cm^−1^, which is typical in FTIR spectra of cheese samples due to their high moisture content [23]; additionally, there were high intensity peaks from CH_2_ and CH_3_ moieties which signify the asymmetrical and symmetrical stretching modes of lipids at 2850–3000 cm^−1^ [24]. Other lipid-associated bands include those from carbonyl stretching at 1742 cm^−1^, that is specifically assigned to triacyclglycerols, a band from ester linkages at 1174 cm^−1^ [23,24,25] and other bands related to C-O stretching around that frequency, and other fat-related bands appearing around 1400–1480 cm^−1^ (from C-H bending modes) [23,25]. Other significant bands are for the amide I and II group vibrations between 1500 and 1700 cm^−1^ [25]. The detailed band assignments are shown in Appendix A.

Raman spectroscopy gathers spectral molecular signatures of the complex food matrixes in a different way from FTIR, returning a high-quality signal for molecules with high polarizability, and being so is a complementary spectroscopic technique to FTIR, which gives a better signal for molecules containing functional groups with a high dipole moment. A typical Raman spectrum is shown in Figure 1b, which is the average of all Raman spectra of all samples measured in this study.

The averaged Raman spectra displays the same features as those found in the literature [26,27]. An intense spectral band between 2800 and 3000 cm^−1^ of CH_2_ and CH_3_ stretch are mainly from fatty acids [28]. Functional group carbonyl mostly from triacylglycerols esters but also from free fatty acids appears at 1742 cm^−1^ [28]. At 1658 cm^−1^, an unsaturated lipid C=C bond stretch arises, overlapping a less intense Amide I band [29]. The Raman fingerprint region is between 400 and 1900 cm^−1^. The Raman fingerprint region is significant because it contains many bands that often overlap. The strongest band in the fingerprint region is from CH_2_ bend mode at 1442 cm^−1^, followed by 1296 cm^−1^ for CH_2_ twist, both for lipids [30,31,32]. Between 1128 and 1063 cm^−1^, three peaks from the C-C stretch are assigned to saturated fatty acids [30,31]. A single peak at 1004 cm^−1^ is assigned to aromatic ring, mainly from phenylalanine [14,28]. Details of the Raman band assignments are given in Appendix A.

### 2.2. Aging Effects within the Same Brand

In this section, we compare the youngest cheeses with the oldest cheeses of the same brand. The samples studied are examples of commercial cheddar cheeses. Between them, there may be variations in feedstock (milk, starter cultures, and other ingredients), cheesemaking processes, and storage conditions. These four brands that comprise the samples are from three different manufacturers, from two different countries, and from two different locations within a country (Table 1). When we compare the aging effects of samples within the same brand, we aim to minimize these variables assuming that they do not vary for products sold under the same brand, and so we designed the study such that we should only observe changes that can be attributed to the cheese ripening process itself. Similar trends are expected between brands, as they essentially comprise the same material at the start of the cheesemaking process: milk, starter, and enzymes.

#### 2.2.1. FTIR Spectral Range

Despite good quality spectra being measured, some spectral regions were found to not contribute to significant variation in the PCA results. In order to obtain the optimal frequency range, three frequency ranges were tested: (i) the full spectra (400–4000 cm^−1^); (ii) a narrower range (400–1800 cm^−1^) similar to that of Subramatrian et al., 2011 [12]; (iii) a range that removes the triglycerol marker at 1742 cm^−1^ (900–1720 cm^−1^). It was found that, in a third of these ranges (900–1720 cm^−1^), the best-differentiated cheddar cheese was based on the maturity level and the manufacturer. Further refinement of the range was undertaken, and the final selected range was 1134–1720 cm^−1^, which comprises most of the FTIR fingerprint region.

A PCA of the IR spectra of cheeses from the four brands (Table 1) was performed to determine whether the spectra can distinguish chemical changes due to aging. The clustering patterns presented in Figure 2a–d clearly show that they can, and similar trends are observed in each of the four plots shown. For all brands, as the ripening time increased, the location of the samples moved to a more positive region of PC 2; however, where there was little difference in the maturity level, there was no discernible change in PC 2 (e.g., GOR 3 and 6 months in Figure 2b; Mainland 8 and 12 months in Figure 2c).

The effect of the maturity level on PC 1 was less distinct, particularly as the data were quite spread-out on the PC 1 axis, meaning that samples from different maturity levels overlapped. In some instances, a small increase in maturity level saw the ranges move a small distance in a positive direction (e.g., GOR 3 and 6 months in Figure 2b; Mainland 8 and 12 months in Figure 2c). In another case, the points moved in a negative direction on PC 1 (Bega 2 and 15 months in Figure 2a).

#### 2.2.2. FTIR PCA

PC 1 is responsible for 85.4% of the total explained variance (TEV), reflecting the fact that the measured spectra have a high degree of similarity. The loadings for PC 1 (Figure 2e) reveal a large positive band at 1162 cm^−1^, assigned to C-O ester linkages [24], a less intense band at 1238 cm^−1^, assigned to C-O groups [15], and a feature at 1464 cm^−1^, from CH_2_ scissoring [15], all of which appear to be lipid-associated bands.

PC 2 (Figure 2f), with total explained variance in 7.1%, has a strong positive band at 1410 cm^−1^, assigned to the = C-H groups of unsaturated fatty acids [33], and 1590 cm^−1^, not assigned to any specific molecular vibrational mode at present. The positive peak at 1590 cm^−1^ is located between the two strongest negative peaks that have their maximum intensity changes at 1546 and 1648 cm^−1^. These are assigned to the amide II (1546 cm^−1^) [15] and amide I (1648 cm^−1^) modes [34]. Together, PC 1 and PC 2 explain 92.5% of the total variance in the FTIR spectra.

#### 2.2.3. FTIR Spectrum Interpretation

In all cases, as the ripening time increased, the location of the samples moves to a more positive region of PC 2, according to the information on loadings plots for PC 1 and PC 2 in Figure 2a–d. PC 1 has a broad positive band with maximum intensity at 1162 cm^−1^, which is assigned to C-O ester linkage bond, but it did not contribute to differentiating between samples with the exception of differentiating between Mainland cheddar samples of 8 and 12 months.

The most relevant contribution to PC 2 comes from the amide regions, where the two strong bands in the negative part of the graphic have their maximum intensity changes at 1546 and 1648 cm^−1^. This region is assigned to the amide II and amide I modes, respectively. Between these two negative bands appears a broad and intense band that is highly positive and with a maximum value at 1590 cm^−1^; however, there is no assignment for this frequency (Appendix A). While no clear band assignment is yet available for this feature, this broad feature is far wider than any vibrational mode, suggesting that it may be generated by a superposition of other changing bands such as the amide I and II bands.

During ripening, proteolysis is the most complex and important set of reactions occurring in cheeses [3]. Caseins and large peptides are hydrolyzed to form small peptides and FAA; so, it is expected that the intensity of the signal from the amide II mode decreases and the signal for amide I increases. The highest contribution in this transition has the highest maturity level, which agrees with the data presented here. Yaman et al., 2022 found the protein region of the IR spectra was relevant for classifying the maturity of their white Turkish cheeses [22], as did de Jesus et al., 2020, in their analysis of the Brazilian artisanal Minas cheese [35], and Fagan et al., 2007, on their homemade Cheddar cheese [36].

#### 2.2.4. Raman Spectral Range

Narrowing the spectral frequency range was shown to be useful to enhance the separation of different samples. This is a common procedure for data analysis as for the FTIR data earlier. An example of bands that may not be useful to highlight spectral differences between samples with different maturity levels is the contribution of lipid bands from 2800 to 3000 cm^−1^. The bands in this region do not change significatively during ripening, as lipolysis acts on the ester linkages, and most of the -CH_2_- backbone from triacylglycerols and free fatty acids remain intact. Thus, this region can be excluded from the data analysis.

In order to obtain the best dataset for performing PCA analysis of the Raman spectra, various ranges were examined: (i) full range (400–3200 cm^−1^), (ii) a narrow range that highlight the fingerprint region (400–1800 cm^−1^), (iii) a narrower range (900–1800 cm^−1^), (iv) another range that removes the carbonyl band from fatty acids (900–1720 cm^−1^), and (v) the range used in the data analysis of FTIR spectra (1134–1720 cm^−1^). Based on the results of those analyses, and further refinement, the most useful spectral range was from 990 to 1800 cm^−1^. All subsequent Raman PCA analysis was performed within this spectral range.

#### 2.2.5. Raman PCA

While differences existed between the brands of cheese, in part due to the variation in the range of maturation times, some general patterns can be observed (Figure 3). In general, on the PC 1 versus PC 2 plots (Figure 3a–d), the cheeses trended from upper left (negative on PC 1, positive on PC 2) to bottom right (positive on PC 1, negative on PC 2). The extent of the change varied from brand to brand, probably due in part to the differences in maturation time. For example, the trend is clearer for Bega cheeses (2–15 months maturation, Figure 3a), but less clear in Mainland brand samples (8–32 months) in Figure 3c. While the range in the reported maturation times was greatest for Mainland cheeses (24 months), the lowest maturation time was relatively long (8 months), so changes during shorter maturation times may be masked.

On the PC 1 versus PC 3 plots, Figure 3e–h, the pattern in the changes is less clear. For example, no pattern in the change is readily apparent for Bega cheeses (Figure 3e); meanwhile, for Mainland cheeses, the lowest maturity samples (8 and 12 months) are in the positive region of PC 3, and the most mature samples (32 months) are located in the negative region of PC 3 (Figure 3g). GOR samples with the lowest maturity (3 and 6 months) tend to be lower on PC 3, with 10 months highest on PC 3, and 18 months lying in an intermediate region (Figure 3f).

#### 2.2.6. Raman Spectra Interpretation

Based on the information from the loadings plots for PC 1, PC 2, and PC 3 (Figure 4), we can determine which bands are responsible for this classification in the score plot. On the PC 2 loadings plot, the region that contributes negatively to the score plot is at a frequency of 1416 cm^−1^ and is assigned to β’-polymorph of fat crystals [37]. This is an interesting result. Despite cheese being a good environment for fat crystallization, to the best of our knowledge, this is the first time that Raman spectroscopy has detected this structure in cheese. Ramel and Marangoni, 2017 characterized the polymorphism of milk fat in processed cheese with differential scanning calorimetry (DSC) and X-ray diffraction (XRD) and found the β’-polymorph is the most abundant [38]. Motoyama et al., 2010, used the Raman band of β’-polymorph at 1417 cm^−1^ to differentiate pork fat from beef fat. Indirect observation of fat crystallinity explained shifts in the CH_2_ bands of 2800–3000 cm^−1^ in cheddar cheese [39]. Aged cheeses tend towards the negative region of PC 2 axis, meaning that the formation of this fat crystal is favored with time. Other bands in Figure 4b, at 1304 and 1262 cm^−1^, are the most positive bands, assigned to CH_2_ twisting (1302 cm^−1^) for saturated fatty acids and = C-H rocking (1260 cm^−1^) for unsaturated fatty acids.

On the loadings plot for PC 3, there are three main bands in the positive region, at 1294 cm^−1^ (CH_2_ twist), 1128, and 1062 cm^−1^ (both from C-C stretching), all assigned to fatty acids. In the negative part of the figure, there are also three bands, at 1432 cm^−1^, 1308 cm^−1^ (these are spectral variations associated with the main peaks at 1440 and 1296 cm^−1^, respectively, both for fatty acids), and 1080 cm^−1^ (C-C stretching for saturated fatty acids). Lipid content changes during cheese ripening through lipolysis and due to the changes in the water content, which decreases with ripening time. These nonpolar molecules do not generally possess high polarizability, but Raman spectroscopy has reliably detected slight differences in signals from the cheddar cheeses studied here. Downey et al., 2005 [40], analyzed cheddar cheese maturation for up to 9 months with NIR. They found that the major contribution to the spectra was from water and lipid bands, and that proteins made no contribution even when loadings were examined up to PC 5. It should be noted that NIR is quite sensitive to water content, while Raman spectroscopy is not.

The same interpretation is generally applicable to cheeses from another manufacturer, GOR. But samples with the two highest levels of maturity are not clearly separated on the score plots comparing PCs 1, 2, and 3 (Figure 3b,f). Going further into the score plots for other PCs, in PC 5, the samples aged for 9–10 months and for 16–18 months are almost totally separated. Appendix A shows the scores plot for PCs 4 × 5. As can be seen on the PC 5 axis, all samples of 9–10 months are located above zero, and just a few of the 16–18-month-old sample measurements are not below the zero value.

On the loadings plot for PC 5 (Appendix A), the high positive region is generated by the band at 1004 cm^−1^, which is assigned to phenylalanine residues, and the most negative feature is at 1654 cm^−1^, representing the amide I vibrational mode. In this way, the intermediary maturation stage makes the phenylalanine contribution more evident. Phenylalanine residues are the preferred target for earlier proteolysis through the residual coagulant and plasmin, resulting in hydrophobic peptides, which are bitter [3]. The very first proteolytic event in cheeses made with proteases (rennet or similar) consists of the hydrolysis of the Phe_105_-Met_106_ bond of κ-casein to form para-κ-casein [3]. Another example is the primary site of chymosin action on α_s1_-casein, the Phe_23_–Phe_24_ bond [41]. Because of this, the Raman signal for this molecular group may be monitoring subtle changes in intensity as function of the ripening time, as revealed here in PC 5.

Eventually, lactic acid bacteria undergo autolysis, releasing their intracellular contents (enzymes and metabolites) into the medium, triggering many enzymatic reactions which produce small peptides and FAA. Studies have previously demonstrated that proteolytic activities were higher with highly autolytic lactococcal starter strains for Cheddar cheese juice within 70 days of ripening [42]. Bacteria can survive for up to 3.5 years in cheese with low levels of metabolism (i.e., with no growth) because of carbohydrate starvation [43]. Secondary bacteria microflora domains appear in the cheese environment after 4 weeks, reaching cfu values of up to 10^9^ [3]. As these samples studied here are produced by 16–18 months and 9–10 months of aging, high levels of autolysis of these bacteria may be happening in between these periods, leading to proteolytic reactions that are different from the earlier stages of cheese maturation, but this hypothesis demands more investigation. De Dea Lindner et al. (2008) showed the ability of natural whey starter to survive for up to 20 months of ripening [44].

In summary, we can see that low maturity Mainland cheeses are concentrated on the top of the positive region of PC 3, the intermediate maturity samples are in the center, around zero of the PC 3, but separated on PC 1, and the highest maturity cheeses are totally separated on the negative region of PC 3. The Cracker Barrel Vintage and Epicure samples are separated along PC 1. All the bands involved in this separation are associated with lipids and have already been discussed in the previous paragraphs.

The overall understanding observed here of aging events based on PCA of Raman spectra is that PC 2 was the most relevant for Bega cheeses, PC 3 for Mainland cheeses, and a diagonal between PC 1 and PC 2 for the GOR and Cracker Barrel samples. The similar behavior of the last two brands may be explained by them both being produced in the same factory with the same feedstock. We note that the cheeses produced in the same factory may vary due to differences in the milk and other ingredients used as well as the exact conditions, even with standardization. This means that some variation in the measured spectra can be expected. However, the results presented in this work showed that the combination of chemometrics with vibrational spectra can reliably identify biochemical differences between samples and due to aging.

### 2.3. Comparing Manufacturer Using Samples with Similar Ripening Time

Samples with similar maturity levels, but from different brands, are compared on the scores plot for FTIR (Figure 5) and Raman spectra (Figure 6), respectively. For the sake of clarity, we organized the data in four levels of maturity. In this section, we seek to investigate the sensitivity of the spectra to variations in these cheddar cheeses due to manufacturer.

The Level 1 group has the shortest time of ripening (2–8 months) and includes Bega Colby, GOR Colby and Tasty, and Mainland Colby cheeses. At this level, only Mainland is clearly distinct from the other level 1 samples for both FTIR and Raman spectra in Figure 5a and Figure 6a,e (lower on the FTIR PC 1 axis and higher on PC 2 axis; on Raman: lower on PC 2 axis and higher on PC 3 axis). Mainland Colby cheese has the highest maturation time of all the level 1 cheeses (8 months compared to 2–6 months for the three other samples), meaning that the separation on the FTIR PC 2 axis was expected, as discussed in Section 2.2.3. A lower PC 1 position on the FTIR axis means there is a less intense contribution from the lactic acid band, which indicates a relevant difference based on the manufacturing process. Note that a characteristic step of the production of Colby cheddars is washing the curd with water at the same amount of whey that was drained [3]. The negative region of the Raman PC 2 axis (Figure 6a) is due to the contribution from unsaturated lipids, so feedstock may explain this difference because this is the only sample from New Zealand in this early stage of aging (level 1). There are three lipid bands in the loadings for Raman PC 3, as discussed in the previous section. It is not expected to see many lipolytic events in the earlier stage of ripening, so this difference may also be linked to the feedstock. The other Cheddar brands overlap with each other on the FTIR and Raman level 1 plots.

The Level 2 aging period ranges from 8 to 12 months. At this level Bega brand cheddars are clustered apart from the other brands on the PCA plot of FTIR spectra (Figure 5b,f). Examining the corresponding Raman spectra plots for Level 2 shows some separation between brands, but with some superposition (Figure 5f and Figure 6b).

The Level 3 category contains cheese aged from 15 to 24 months. With FTIR data, it is impossible to distinguish any brand here except for Cracker Barrel, which has data points at slightly more negative PC 2 values than the others, as can be seen in Figure 5c. When the corresponding Raman plots for this maturation level are examined, distinct clusters for each brand are more obvious and partially distinguishable from each other, as is shown in Figure 6c,g.

Finally, the Level 4 maturation category contains the most mature cheeses, and the two samples studied here were the Mainland Epicure and Cracker Barrel Epicure brands. Both cheddar cheeses have an aging time of 32 months. Analysis of the FTIR and Raman spectra shows the two cheeses are well-separated on PC 2 for FTIR (Figure 5d) and PC 3 for Raman spectra (Figure 6h).

The comparison between brands with similar maturity time presented here is complicated by the fact that the aging time of each sample in the four categories is not necessarily the same between different manufacturers. The compositional content of the samples ranges from 22.6 to 27% of protein, and from 34 to 37% of fat, according to the label information, presented in Appendix A. Because the compositional content is quite identical, and the ripening times are very similar, the samples in these group tend to cluster. Despite that, the spectral differences identified here cannot be only due to the effects of the differences in aging time on oxidation, dehydration, proteolysis, and lipolysis because cheeses of the same brand tend to cluster, especially in their Raman spectra. For example, in our highest maturation stage group, Level 4, the samples with the very same aging time are totally separated in the score plots, which indicates consistency in the cheesemaking process (e.g., processing or maturation temperature) or feedstock (milk, starter). Such consistency is ensured by manufacturers throughout the ripening process for these commercially available cheddar cheeses.

### 2.4. Features Detected by FTIR and Raman Spectroscopies

This work demonstrates the ability of FTIR and Raman spectroscopy to classify commercially available cheddar cheeses based on their maturity level and based on their manufacturer. The main chemical features responsible for this classification are the amide I and II bands in the FTIR spectra; meanwhile, the lipid-associated bands were the most relevant in the Raman spectra. The amide l mode makes a small, yet observable, contribution to the Raman spectra of long-matured cheese, revealing that a relevant proteolytic event could be detected between 10 and 18 months of maturation. The main features detected for each technique are summarized in Table 2.

## 3. Materials and Methods

### 3.1. Samples

Commercially available cheddar cheese samples from four different commercial brands and with differing levels of maturity were purchased from local retailers. There are a number of common maturity level descriptors used in the Australian market; however, the number of months of maturation varied from brand to brand (Table 1) and not all commercial brands had all maturity levels. In all, there were 14 different types of cheese (brand and maturity levels; blocks of 250–500 g) and the samples were collected in triplicate, giving a total of 42 samples. After purchase, the samples were stored at 4 °C until they were required for analysis. Before spectra collection, the samples were allowed to warm up to room temperature for 1 h. The inner part of the cheese blocks was chosen to collect the spectra, to avoid changes in the composition of the external part of the samples due to oxidation, dehydration, and photodecomposition. Half a centimeter was cut off from the borders. Then, flat slices were cut off near the center of the blocks and used for analysis.

### 3.2. Spectral Sampling Acquisition

#### 3.2.1. FTIR

A Perkin Elmer FT-IR spectrometer was used to collect the FTIR spectra. It was equipped with the ATR accessory (Pike Technologies, GladiATR, Madison, WI, USA). For each of the samples, 20 FTIR spectra were collected at random positions to account for samples heterogeneity with respect to the dimensions of the evanescent field. The ATR accessory was used to press the sample on to the crystal to allow reasonable contact so that good quality spectra could be collected. A total of 32 accumulations per spectrum were collected, within a spectral range from 400 to 4000 cm^−1^ and a spectral resolution of 4 cm^−1^. After acquisition, the spectra were normalized before chemometrics by center data to have mean zero and scale data to attain a standard deviation equal to 1.

#### 3.2.2. Raman

A PerkinElmer Raman Station 400 spectrometer was used to collect the Raman spectra. The detector was cooled to −50 °C. The exposure time was 5 s and there were 12 accumulations, giving an acquisition time of 1 min per spectrum. The excitation laser operated at 785 nm and 250 mW of power. The spectral range measured was 400–3200 cm^−1^ at a resolution of 2 cm^−1^. Before the collection of Raman spectra, background scans were collected with the same characteristic of the spectra, i.e., 5 s and 12 accumulations. For each sample, 20 spectra were collected at random locations on the surface. The spectra were pre-treated before chemometrics by removing cosmic rays, and then a baseline correction was performed to correct the fluorescence contribution. Next, a smoothing function was applied using the first derivative of the Savitzky–Golay smoothing method with a window of 5 data points. Lastly, a normalization was conducted in the same way as for FTIR spectra.

### 3.3. Multivariate Analysis

Pre-processing and multivariate analysis of the spectra were carried out using MATLAB R2021b (Mathworks Inc., Natick, MA, USA). Principal component analysis (PCA) is an exploratory data analysis technique that aims to reduce the amount of data to a few relevant principal components (PCs) obtained from the linear combinations of old variables from the dataset. [47] The first PC explains the largest amount of variation (expressed as a percentage). The second PC explains the second largest amount of explained variance in the original dataset, and so on. The numerical values associated with PCs of each sample are the score values, and they explain how the samples are related to each other. In the process of obtaining the PCs, the original dataset is projected in the hyperspace, generating a series of vectors that are the Eigen vectors of the set, known as loading vectors. The numeric values of the loading vectors are the loading values, and they explain how the variables are related to each other. PCA is probably the most widespread multivariate chemometric technique [48] and can quickly identify the most significant changes when comparing identical spectra from several samples.

In this work, the results presented are from data analysis using the spectral ranges stated in Section 2.2.1 for FTIR (1134–1720 cm^−1^), and in Section 2.2.4 for Raman (990–1800 cm^−1^). Each point in the score plots correspond to an individual spectrum. No average was performed. In all cases, the data analysis was performed for the whole dataset, but only the desired group is presented in each figure.

## 4. Conclusions

Our PCA analysis of the FTIR spectra showed clear separation among cheddar cheeses based on the maturity level; meanwhile, our PCA analysis of the Raman spectra gave a better classification in terms of the specific manufacturer for cheeses with similar maturities. An important feature that allowed this classification was the optimization of the specific spectral ranges for the data analysis. The data shown here demonstrate the future potential of these two spectroscopic techniques in routine application in industry for assessing the degree of maturation; such an approach will potentially allow the earlier release of a product while maintaining its quality, providing a commercial advantage. The techniques also have potential value for regulatory agencies, to allow rapid detection of counterfeit products. This study may find applications in other dairy and food products where there are chemical transformations associated with aging.

While near-infrared analysis has been routinely applied for cheese measurements in the dairy industry [49], our work presented here shows that other vibrational spectroscopic techniques can be a valuable extension of this approach.

The degree of heterogeneity in these samples increased with the cheese’s ripening, and this was particularly evident through PCA analysis of the FTIR spectra, illustrating that cheese maturation is a heterogeneous process, occurring at different rates across and throughout the cheese matrix. Further investigations based upon this study may reveal more details about the heterogeneity of cheese maturation and ripening.

## Figures and Tables

**Figure 1 molecules-28-08051-f001:**
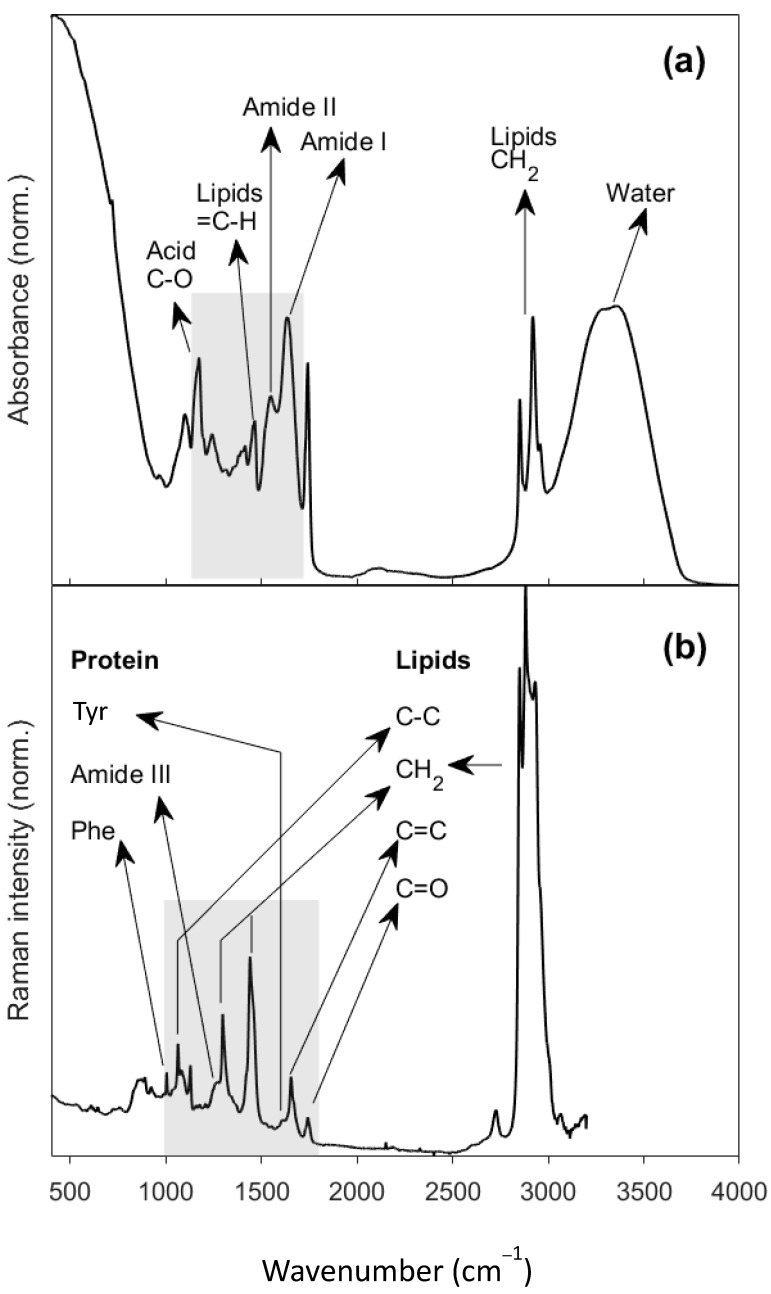
(**a**) Averaged FTIR spectrum of cheddar cheeses using an attenuated total reflectance accessory; (**b**) averaged Raman spectrum of cheddar cheeses. The grey rectangles show the regions of the spectra used for data analysis.

**Figure 2 molecules-28-08051-f002:**
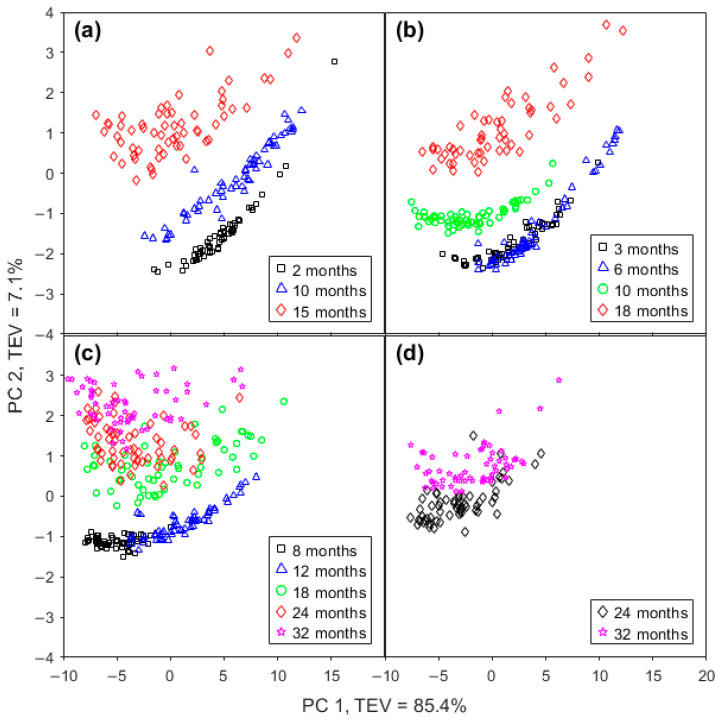
Score plots of PC 1 and PC 2 from FTIR spectra of (**a**) Bega, (**b**) Great Ocean Road, (**c**) Mainland, and (**d**) Cracker-Barrel cheddar cheese. (**e**) Loadings plot of PC 1 and (**f**) loadings plot of PC 2 of PCA from FTIR spectra. TEV means total explained variance. Dashed line represents the averaged FTIR spectrum.

**Figure 3 molecules-28-08051-f003:**
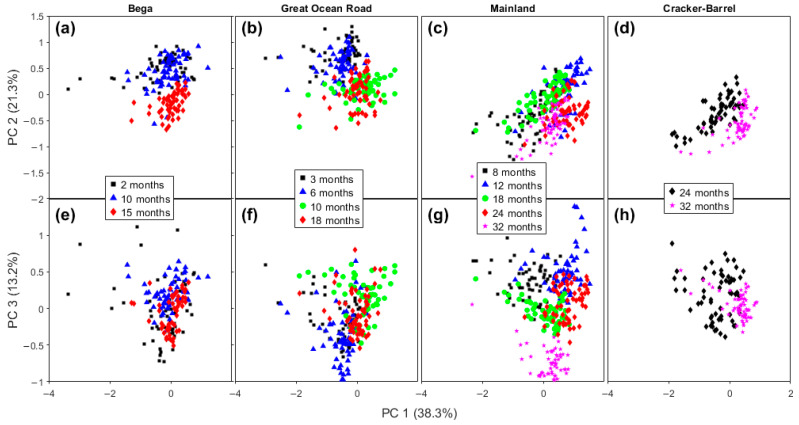
Score plots of PC 1 × PC 2 (**a**–**d**), and PC 1 × PC 3 (**e**–**h**) of Bega, Great Ocean Road, Mainland, and Cracker-Barrel cheddar cheeses from PCA analysis of their Raman spectra.

**Figure 4 molecules-28-08051-f004:**
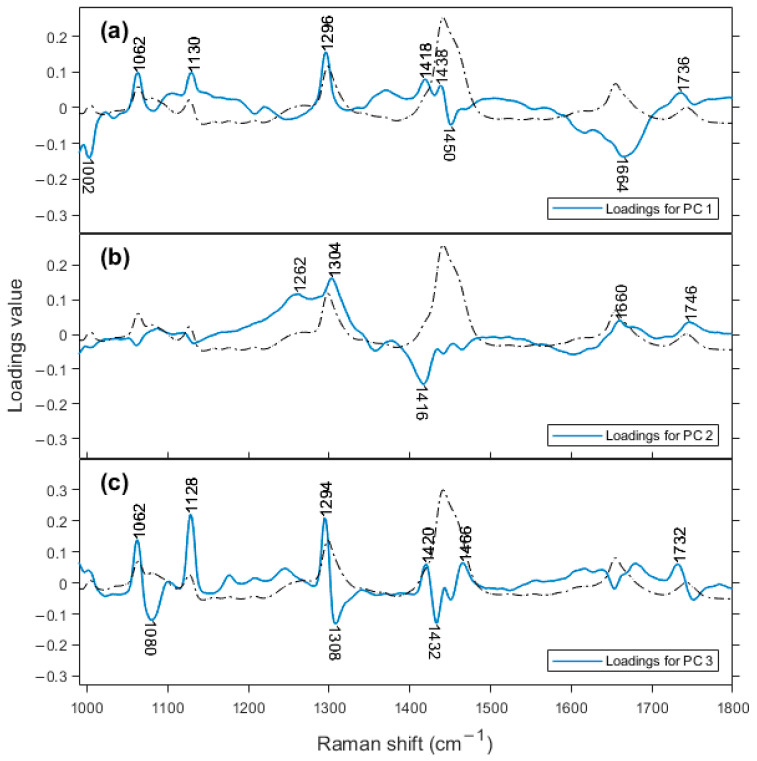
Loadings plots for (**a**) PC 1, (**b**) PC 2, and (**c**) PC 3. Dashed line represents the averaged Raman spectrum.

**Figure 5 molecules-28-08051-f005:**
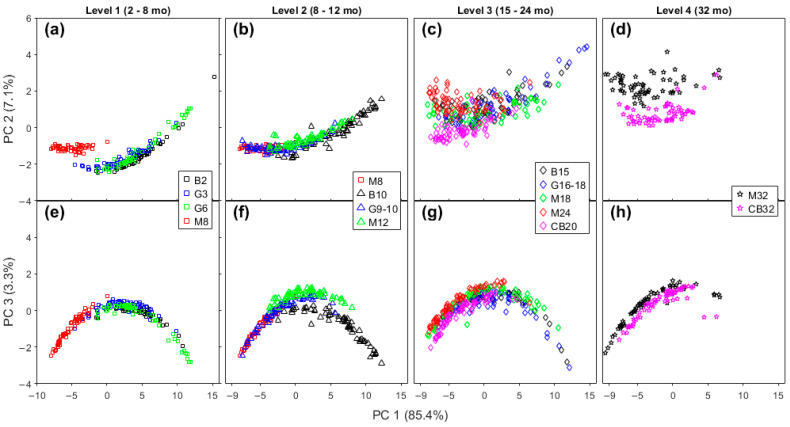
Scores plots of FTIR spectra comparing four brands of cheddar cheese in a close range of aging time. Score plots of PC 1 × PC 2 (**a**–**d**), and PC 1 × PC 3 (**e**–**h**) of Level 1 (**a**,**e**), Level 2 (**b**,**f**), Level 3 (**c**,**g**), and Level 4 (**d**,**h**). Legend: B—Bega; G—Great Ocean Road; M—Mainland; and CB—Cracker Barrel.

**Figure 6 molecules-28-08051-f006:**
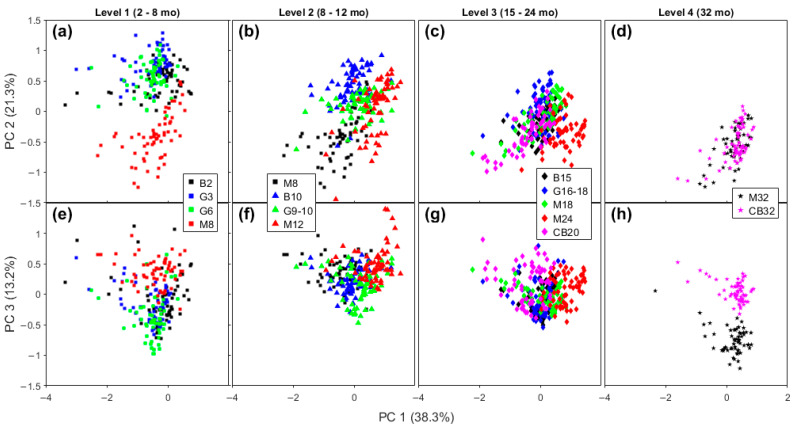
Scores plot of Raman spectra comparing four brands of Cheddar cheese in a close range of aging time. Score plots of PC 1 × PC 2 (**a**–**d**), and PC 1 × PC 3 (**e**–**h**) of Level 1 (**a**,**e**), Level 2 (**b**,**f**), Level 3 (**c**,**g**), and Level 4 (**d**,**h**). Legend: B—Bega; G—Great Ocean Road; M—Mainland; and CB—Cracker Barrel.

**Table 1 molecules-28-08051-t001:** Cheese samples by commercial brand and maturity level.

Brand (Company; Location)	Maturity Level (Months) ^1^				
	Colby	Tasty	Extra Tasty	Vintage/ Extra Sharp	Epicure
Bega (Bega; New South Wales)	2	10	-	15	-
Great Ocean Road (GOR) (Saputo; Victoria)	3	6	9–10	16–18	-
Cracker Barrel (CB) (Saputo; Victoria)	-	-	-	20	32
Mainland (Fonterra; New Zealand)	8	12	18	24	32

^1^ According to the information provided by the manufacturer.

**Table 2 molecules-28-08051-t002:** Molecular groups identified by each technique. Inside parenthesis indicates when the band contributed to the principal component analysis.

	FTIR (cm^−1^)	Raman (cm^−1^)	Ref
Moisture	Intense 3000–3600	Not detected in this work	[23]
Lipids			
Saturated fatty acids	2800–3000, 1464, 1394, 1354	2800–3000, 1440 (PC 1, PC 2, PC 3), 1296 (PC 2, PC 3), 1128 (PC 3), 1080, 1063 (PC 3)	[15,24,28,30,31,33]
Unsaturated fatty acids	1414	1266 (PC 2), 1421 (PC 2), 959	[29,30,33]
Ester linkage	1174 (PC 1)		[24]
Carbonyl (most from triacylglycerols)	1742	1742	[15,24,28]
Fat crystals	720	1416 (PC 2)	[37,45]
Proteins			
Amide bands	1548 (PC 2), 1636 (PC 2)	1658 (PC 5)	[15,29,34]
Specific AA	Not detected	Cystine (540), Tyrosine (833, 1616), Tryptophan (760, 889, 1370), Phenylalanine (1004) (PC 1 and PC 5)	[14,28,29]
Other molecular groups			
Lactic Acid	1098 (PC 1)	Not detected	[15]
Carotenoids	Not evident	1158, 1526.	[31]
Phospholipids	Not evident	846, 852, 868.	[28,46]

## Data Availability

Data are contained within the article and Appendix A.

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
