# Peer review of "Identifying Chemical Differences in Cheddar Cheese Based on Maturity Level and Manufacturer Using Vibrational Spectroscopy and Chemometrics"

_molecules, 2023, doi:10.3390/molecules28248051_

Round 1
Reviewer 1 Report
Comments and Suggestions for Authors
What is the accuracy of using FTIR and Raman spectroscopies for detect the effects of ageing in commercial cheddar cheeses?
The last paragraph in the introduction section that represent the novelty of work should be rewrite.
Will this study useful for using FTIR and Raman spectroscopies for detection the aging effect in dairy products?
Comments on the Quality of English LanguageThere are many mistakes in English language in whole MS. Revise the MS carefully
Author Response
Reviewer 1
What is the accuracy of using FTIR and Raman spectroscopies for detect the effects of ageing in commercial cheddar cheeses?
To quantify the accuracy of the classification of the aging classes was not the objective of this work. Our aim was identifying changes based on ageing and manufacturer, as stated on the title, not to calculate the accuracy of these techniques. In this work, we have focused on interpreting the chemical aspects revealed in the loadings plots. PCA is the simplest chemometric and it is highly suitable for the subject of this study. However, if PCA is not sufficient to separate classes, a more sophisticated chemometric technique may be used. For the samples studied in this work, we estimate near 100% accuracy from the FTIR spectra, because the 3D cluster plot showed total separation of the classes (data not shown). Please also see our response to Reviewer 3 who made an identical query.
The last paragraph in the introduction section that represents the novelty of work should be rewritten
The last paragraph in the introduction section has been rewritten as follows:
“Our research aims to identify chemical changes in commercial cheddar cheese due to aging and due to manufacturer. Non-destructive techniques FTIR and Raman were applied to, and the spectral variations were identified by principal component analysis (PCA). Ripening events, manufacturing conditions and feedstock lead to different transformations in cheese. Note that in this manuscript we use the term ‘feedstock’ to include all the ingredients that play a role in cheddar maturation: milk, LAB, rennet and salt.”
Will this study useful for using FTIR and Raman spectroscopies for detection the aging effect in dairy products?
We believe these methods will be useful for studying the aging effect in dairy products in general. While it is not clear if common aging effects other than those induced by e.g. the bacteria in yoghurt would be highlighted in the same way as found for cheddar cheeses here, the methodology we have developed is broadly applicable to different sample types and is sensitive to any of the chemical associated by aging, namely oxidation, dehydration, proteolysis and lipolysis.
We have updated the conclusion to reflect this, by adding the following text:
“This study may find applications on other dairies and food products were there are chemical transformations associated with aging.”
Reviewer 2 Report
Comments and Suggestions for Authors
General comments: This paper covers the application of mid infrared and Raman spectroscopy to Cheddar cheese as it matures. The interpretation of the spectral data gives insights into the chemical processes that occur as cheese undergoes proteolysis. The paper would benefit from some additional information especially regards cheese composition and ageing conditions and how this could impact the observed spectral differences.
Specific comments:
Lines 30/63/73/190/286/329/338/362/364/427: Capital “Cheddar”
Line 61: Correction “as well as testing”
Line 75: Addition “relevant spectral bands”
Lines 76/117/306/329/360: Clarification suggest change “feedstock” to “ingredient raw milk”
Figure 1a/b: Is it possible to show the variation in spectral data collected along with the average spectra as shown?
Line 128: Suggest reword “found not to contribute to significant variation in the PCA results”
Line 128: Please add in reference to Figure 2 a-d PCA results and clarify what spectral data range was used in the PCA plots shown here. Also clarify what each data point in the PCA plot represents, is it an average of the triplicates measured or is every individual point plotted? Please also clarify what spectral data are the loadings pertaining too – all or selected cheese ages?
Lines 280-282: Explain further or reference why proteolysis enhances amide I signal
Line 306: Clarify that cheeses produced in the same factory may vary given raw milk, even though standardised to consistent fat/protein ratios can still exhibit differences due to the seasonal variation of raw milk. In addition rinsing conditions may vary between manufacturers.
Figure 5: In order to draw strong conclusions from PCA of individual cheeses from different manufacturers it is important to determine that the major variations explained are not due to compositional differences. If possible, can you note the composition of each cheese so as to rule this out?
Line 359: Not sure of logic that samples different in second PC (level 4 maturation) have consistent cheesemaking and raw milk, doesn’t this suggest they are different (and see earlier comment regards composition and ripening conditions)
Line 375: Samples. The authors must include detailed on how the cheese samples were subsampled to give material that was subjected to spectroscopy. Blocks of cheese are not homogeneous and its relevant to readers to understand where and how the subsamples were extracted from then purchased blocks.
Line 391: The authors should note that the use of ATR for measuring cheese may not give fully representative spectra given the depth of penetration of the evanescent wave into the cheese may be less that some of the microstructural components of the cheese such as fat globules.
Line 431: Correction “data shown”.
Line 433: Suggest the authors note that near infrared analysis is more routinely applied to cheese measurement in industry, so perhaps a reference to this would afford useful context (Alinaghi M. ND, Singh N., Höjer A., Saedén K.H., Trygg J. Near-infrared hyperspectral image analysis for monitoring the cheese-ripening process. Journal of Dairy Science. 2023;106:7407-18.)
Comments on the Quality of English Language
As above
Author Response
REVIEWER 2
General comments: This paper covers the application of mid infrared and Raman spectroscopy to Cheddar cheese as it matures. The interpretation of the spectral data gives insights into the chemical processes that occur as cheese undergoes proteolysis. The paper would benefit from some additional information especially regards cheese composition and ageing conditions and how this could impact the observed spectral differences.
Thank you for your comment. A table (Table S1) with cheese composition has been added to the SI file. The following comments about cheese composition have been added to the last paragraph of the section ‘2.3. Comparing manufacturer using samples with similar ripening time’:
“The compositional content of the samples ranges from 22.6 to 27% of protein, and from 34 to 37% of fat, according to the label information, present in Table S3. Because the compositional content is quite identical, and the ripening time is very near to each other, the samples in these group tend to cluster to each other.”
As these were commercial samples obtained at supermarkets, unfortunately we do not have details about the aging conditions.
Specific comments:
Lines 30/63/73/190/286/329/338/362/364/427: Capital "Cheddar"
These have been corrected.
Line 61: Correction "as well as testing"
This has been corrected.
Line 75: Addition "relevant spectral bands”
This paragraph has been rewritten, as requested by Reviewer 1.
Lines 76/117/306/329/360: Clarification suggest change "feedstock" to "ingredient raw milk”.
Despite the major feedstock of the cheese being milk, the others namely LAB, rennet and salt are also important raw ingredients. In the manuscript we refer to feedstock as milk, LAB, salt and rennet because all play significant roles in the development of the maturation process.
We have added the following text in the last paragraph of the introduction, that has been rewritten as request from Reviewer 1:
“In this manuscript we use the term ‘feedstock’ to include all the ingredients that play a role in cheddar maturation: milk, LAB, rennet and salt.”
Figure 1a/b: ls it possible to show the variation in spectral data collected along with the average spectra as shown?
The spectral variation in the raw spectra is tiny and could confuse the reader. An example of spectral variation has been included in Supplementary Information as Figure S1. The following text has been added to the first paragraph of the Results and Discussion section:
“An example of spectral variation has been included in Supplementary Information in Figure S1.”
Line 128: Suggest reword "found not to contribute to significant variation in the PCA results'
This change has been made.
Line 128: Please add in reference to Figure 2 a-d PCA results and clarify what spectral data range was used in the PCA plots shown here. Also clarify what each data point in the PCA plot represents, is it an average of the triplicates measured or is every individual point plotted? Please also clarify what spectral data are the loadings pertaining too - all or selected cheese ages?
Thank you for your question. The follow paragraph is now included in the end of the section 3.3 Multivariate Analysis:
“In this work, the results presented are from data analysis using the spectral ranges stated in sections 2.2.1 for FTIR (1134 – 1720 cm-1), and in section 2.2.4 for Raman (990 – 1800 cm-1). Each point in the scores plots correspond to an individual spectrum. No averaging was performed. In all cases, the data analysis was performed over the whole data set, but only the desired group is presented in each figure.”
Lines 280-282: Explain further or reference why proteolysis enhances amide I signal
We have deleted this paragraph and reference 40.
Line 306: Clarify that cheeses produced in the same factory may vary given raw milk, even though standardised to consistent fat/protein ratios can still exhibit differences due to the seasonal variation of raw milk. In addition rinsing conditions may vary between manufacturers
The reviewer is correct. We have added the following text at the end of the referred line:
“We note that cheeses produced in the same factory may vary due to differences in the milk and other ingredients used as well as the exact conditions, even with standardization. This will lead to some variation in the measured spectra being expected. However, the results presented in this work showed that the combination of chemometrics with vibrational spectra can reliably identify biochemical differences between samples and due to aging.”
Figure 5: In order to draw strong conclusions from PCA of individual cheeses from different manufacturers it is important to determine that the major variations explained are not due to compositional differences. If possible, can you note the composition of each cheese so as to rule this out?
A table with the composition of each cheese is now included in the SI as Table S3. Also, a text has been added to the last paragraph of the section ‘2.3. Comparing manufacturer using samples with similar ripening time’ as answered on the Reviewer’s first comment, above.
Line 359: Not sure of logic that samples different in second PC (level 4 maturation) have consistent cheesemaking and raw milk doesn't this suggest they are different (and see earlier comment regards composition and ripening conditions)
Line 375: Samples. The authors must include detailed on how the cheese samples were subsampled to give material that was subjected to spectroscopy. Blocks of cheese are not homogeneous and its relevant to readers to understand where and how the subsamples were extracted from then purchased
Reviewer 3 also asked more details about the sample preparation. Please see the answer given there.
Line 391: The authors should note that the use of ATR for measuring cheese may not give fully representative spectra given the depth of penetration of the evanescent wave into the cheese may be less that some of the microstructural components of the cheese such as fat globules.
Yes, we agree with the reviewer, that the heterogeneity of cheese samples with respect to the evanescent field is an important consideration, which is why we collected 20 replicate spectra for each sample in order to obtain a reliable average.
We have modified the second sentence in the section 3.2.1 FTIR as follows:
“For each of the samples, 20 FTIR spectra were collected at random positions to account for samples heterogeneity with respect to the dimensions of the evanescent field.”
Line 431: Correction "data shown"
This has been corrected.
Line 433: Suggest the authors note that near infrared analysis is more routinely applied to cheese measurement in industry, so perhaps a reference to this would afford useful context (Alinaghi M. ND, Singh N., Hõjer A., Saedén K.H., Trygg J. Near-infrared hyperspectral image analysis for monitoring the cheese-ripening process. Journal of Dairy Science. 2023;106:7407-18.)
We thank the reviewer for this suggestion, and we have added the following text in the Conclusion:
“While Near-Infrared analysis has been routinely applied to cheese measurements in the dairy industry [49], our work presented here shows that other vibrational spectroscopic techniques can be a valuable extension of this approach.”
And we have added a new reference in the reference list.
- Alinaghi M. ND, Singh N., Hõjer A., Saedén K.H., Trygg J. Near-infrared hyperspectral image analysis for monitoring the cheese-ripening process. Journal of Dairy Science. 2023;106:7407-18.)
Reviewer 3 Report
Comments and Suggestions for Authors
Dear Authors,
The paper "Identifying chemical differences in cheddar cheese based on maturity level and manufacturer using vibrational spectroscopy and chemometrics," primarily focuses on the use of spectroscopic techniques to differentiate cheddar cheese based on various factors like maturity level and manufacturer. The manuscript has a significance of content-making suitable for publication. However, it needs some improvements. I hope my comments help:
To enhance the clarity and interpretability of Figure 1, I suggest rearranging the presentation of the two spectra by positioning them vertically, with one above the other. This layout would aid in a more straightforward comparison between them. Additionally, it would be beneficial to mark the specific bands of interest. This can be achieved by using arrows or lines, which would directly guide the reader’s attention to these key features and facilitate a more efficient and focused analysis.
The current placement of Table 2 on page 13 may contribute to a somewhat challenging reading experience. To improve the flow and accessibility of the manuscript, I suggest reconsidering the positioning of Table 2. Aligning it more closely with the relevant text sections could enhance the reader's ability to integrate the table's information with the corresponding narrative smoothly, thus alleviating any potential disruption in the reading process.
I noticed an isolated question mark ("?") on page 4, line 136, and it was unclear whether this was an inadvertent typographical error or an intentional part of the text. Could you please clarify this? If it is a typo, removing or correcting it would contribute to the manuscript's overall clarity and professionalism.
On page 4, line 137, the usage of the word 'clearly' to describe the separation of PCA scores might not be entirely precise. There are instances of score overlap, which suggests that the separation isn't as distinct as indicated. Could the authors elaborate on how they quantified the level of separation for these PCA scores?
Furthermore, the term 'mo' used in Figure 1 is ambiguous. It seems to imply 'month,' but this is only made clear later in the text. For better clarity and ease of understanding, I suggest explicitly defining 'mo' at its first occurrence or consistently using the full term 'month.' This would significantly enhance the readability and comprehension of the manuscript in its early sections.
In Section 2.2.2, which covers FTIR PCA, the explanation provided for the principal component (PC) loadings is notably precise and detailed. It would be beneficial if the authors could incorporate additional references that support and contextualize their explanation to strengthen this section further. Including relevant literature would enhance the credibility of the analysis and provide readers with resources for deeper understanding or further investigation.
In the 'Materials and Methods' section, explicitly following subsection 3.1, there appears to be a crucial omission concerning the sample preparation process. The manuscript does not provide detailed information on how the samples were prepared for the study. Including a dedicated subsection on sample preparation would be highly beneficial for readers, especially those who may wish to replicate your study. Detailed sample preparation instructions are essential for reproducibility and ensuring the experimental methodology's integrity.
In Section 3.2, currently titled 'Vibrational Analysis,' I suggest revising the title to 'Spectral Sampling Acquisition.' This alternative title more accurately reflects the content of the section, which primarily focuses on the methods and procedures for acquiring spectral data. Such a change would enhance the clarity and precision of the section's heading, aligning it more closely with the described methodologies.
In the section '2.3. Comparing samples with similar ripening times,' I suggest exploring the potential of evaluating samples with more extended ripening times. For instance, comparing samples aged 2 (or 3) months with those aged 15 (or 18, 24, 32) months could provide insightful data on the method's applicability over longer durations. This suggestion stems from observing the score plots in Figures 2 and 3, which indicate that the method may be well-suited for distinguishing samples over a broader time range, not just within shorter intervals.
Would it be possible for the authors to extend their statistical analysis to include these longer ripening times? Such an investigation could significantly enhance the study's scope and demonstrate the method's versatility in different aging contexts. This addition might reveal more profound insights into the spectral changes occurring over an extended period and potentially broaden the method's applicability.
I commend the authors for their diligent work and valuable contributions to the field. The observations and suggestions provided above are intended to enhance the manuscript's clarity, precision, and overall impact. Implementing these modifications could significantly improve the reader's comprehension and the study's replicability, thereby extending its reach and utility within the scientific community.
Author Response
REVIEWER 3
The paper "ldentifying chemical differences in cheddar cheese based on maturity level and manufacturer using vibrational spectroscopy and chemometrics," primarily focuses on the use of spectroscopic techniques to differentiate cheddar cheese based on various factors like maturity level and manufacturer. The manuscript has a significance of content-making suitable for publication. However, it needs some improvements. I hope my comments help:
To enhance the clarity and interpretability of Figure 1, I suggest rearranging the presentation of the two spectra by positioning them vertically, with one above the other. This layout would aid in a more straightforward comparison between them. Additionally, it would be beneficial to mark the specific bands of interest. This can be achieved by using arrows or lines, which would directly guide the reader's attention to these key features and facilitate a more efficient and focused analysis
Thank you for the suggestions. Figure 1 has been updated accordingly.
The current placement of Table 2 on page 13 may contribute to a somewhat challenging reading experience. To improve the flow and accessibility of the manuscript, I suggest reconsidering the positioning of Table 2. Aligning it more closely with the relevant text sections could enhance the reader's ability to integrate the table's information with the corresponding narrative smoothly thus alleviating any potential disruption in the reading process
We have made this change.
I noticed an isolated question mark ("?") on page 4, line 136, and it was unclear whether this was an inadvertent typographical error or an intentional part of the text. Could you please clarify this? If it is a typo, removing or correcting it would contribute to the manuscript's overall clarity and professionalism
We have removed this.
On page 4, line 137, the usage of the word 'clearly' to describe the separation of PCA scores might not be entirely precise. There are instances of score overlap, which suggests that the separation isn't as distinct as indicated. Could the authors elaborate on how they quantified the level of separation for these PCA scores?
Despite there not being a total separation, the same trend can be observed. Therefore, we have changed “clearly” to “reasonably well”. This is a qualitative rather than quantitative assessment of separation of the clusters in the PCA plots, we have not used any metric to quantify the separation.
Furthermore, the term 'mo' used in Figure 1 is ambiguous. It seems to imply 'month,' but this is only made clear later in the text. For better clarity and ease of understanding, I suggest explicitly defining 'mo' at its first occurrence or consistently using the full term 'month.' This would significantly enhance the readability and comprehension of the manuscript in its early sections.
All Figures have been updated.
In Section 2.2.2, which covers FTIR PCA, the explanation provided for the principal component (PC) loadings is notably precise and detailed. It would be beneficial if the authors could incorporate additional references that support and contextualize their explanation to strengthen this section further. Including relevant literature would enhance the credibility of the analysis and provide readers with resources for deeper understanding or further investigation
Thank you for your comment. Four different references have been added to this section, cited six times.
In the 'Materials and Methods' section, explicitly following subsection 3.1, there appears to be a crucial omission concerning the sample preparation process. The manuscript does not provide detailed information on how the samples were prepared for the study. Including a dedicated subsection on sample preparation would be highly beneficial for readers especially those who may wish to replicate your study. Detailed sample preparation instructions are essential for reproducibility and ensuring the experimental methodology's integrity
Thank you for your comment. More details in the experimental section were requested for Reviewer 2 as well. The sample preparation was very simple. We have included the text bellow in the section 3.1 Samples:
“Before spectra collection, the samples were allowed to warm uptot room temperature for 1h. The inner part of the cheese blocks was chosen to collect the spectra, to avoid changes in the composition of the external part of the samples due to oxidation, dehydration, and photodecomposition. Half a centimetre was cut off from the external borders. Then, flat slices were cut off near the centre of the blocks and used for analysis.”
In Section 3.2, currently titled 'Vibrational Analysis,' I suggest revising the title to 'Spectral Sampling Acquisition.' This alternative title more accurately reflects the content of the section, which primarily focuses on the methods and procedures for acquiring spectral data. Such a change would enhance the clarity and precision of the section's heading, aligning it more closely with the described methodologies
The title of the referred section has been changed according to the suggestion of the Reviewer.
In the section '2.3. Comparing samples with similar ripening times,' I suggest exploring the potential of evaluating samples with more extended ripening times. For instance, comparing samples aged 2 (or 3) months with those aged 15 (or 18, 24, 32) months could provide insightful data on the method's applicability over longer durations. This suggestion stems from observing the score plots in Figures 2 and 3, which indicate that the method may be well-suited for distinguishing samples over a broader time range, not just within shorter intervals
All data analysis included the whole data set. Comparations have been made between all So, the oldest cheeses are included in and discussed. Reviewer 2 wondered which part of the data set were used. We have improved these details in the experimental section, please refer to our responses to Reviewer 2.
Would it be possible for the authors to extend their statistical analysis to include these longer ripening times? Such an investigation could significantly enhance the study's scope and demonstrate the method's versatility in different aging contexts. This addition might reveal more profound insights into the spectral changes occurring over an extended period and potentially broaden the method's applicability
Thank for your comment. The analyses that have been done included the youngest (Colby – 2-3 months) and the oldest Epicure (32 months) samples as well. We have added the following text in section ‘2.2 Aging effects within the same brand’ to further clarify this point:
“In this section, we compare the effects of aging of cheddar cheeses of the same brand..”
I commend the authors for their diligent work and valuable contributions to the field. The observations and suggestions provided above are intended to enhance the manuscript's clarity precision, and overall impact. Implementing these modifications could significantly improve the reader's comprehension and the study's replicability, thereby extending its reach and utility within the scientific community
We do appreciate the Reviewer’s valuable comments and suggestions. Thank you very much.
Round 2
Reviewer 3 Report
Comments and Suggestions for Authors
Dear Authors,
I have reviewed the document, and all the comments have been addressed.
I hope this work reaches a high impact.
Good luck!